# Aesthetic Preference of Timber Joints in Architectural Products

**Blair Kuys *** and **Mozammel Mridha**

School of Design and Architecture, Swinburne University of Technology, Melbourne 3122, Australia;
mmridha@swin.edu.au
* Correspondence: bkuys@swin.edu.au

**Abstract:** This study investigates how Australian consumers' aesthetic preferences for timber joints in architectural products are influenced by their sociodemographic characteristics and the visual appearance of the joints. Visual appearance in architecture and product design is a vital factor in consumer response and success of a product; however, designed items are often created without aesthetic research rigour to better understand user acceptance. We see this as an opportunity for greater penetration of aesthetics for designed products and, in this instance, contemporary architecture. We provide extensive literature defining aesthetics and outline the theoretical framework for experimental computer-generated visual stimuli. An online survey was conducted with 114 participants, who rated five timber joints on seven visual appearance attributes. The findings reveal that Joint 1 (angular) and Joint 5 (curved) were the most preferred joints. Employment status was the only sociodemographic factor that significantly affected the aesthetic preference. The findings of the study were used to inform design decisions for building a pagoda in a cemetery in Melbourne. The study contributes to the literature on aesthetics and design by providing empirical evidence on consumer preferences for architectural products. The study also suggests an opportunity to bridge aesthetics with sustainability, as timber is a sustainable material that can be designed to resonate with consumers' aesthetic sensibilities while adhering to environmental principles.

**Keywords:** aesthetic preference; timber joints; architectural products; visual appearance; sociodemographic factors

## 1. Introduction

Aesthetics is an important factor in product design, as it influences consumer response and product success [1,2]. Aesthetics refers to the subjective inner experience, judgment, and evaluation of humans toward objects that are perceived as beautiful or pleasing [3]. Aesthetic preference is a specific type of aesthetic response that reflects the degree of liking or attraction toward an object [4]. Aesthetic preference can be influenced by various factors, such as the visual appearance of the object, the personal attributes of the consumer, and the cognitive and affective processes involved in the evaluation [5,6]. One domain where aesthetics plays a crucial role is architecture, as it shapes the built environment and affects human well-being [7]. Architectural products, such as buildings, structures, or furniture, have both functional and aesthetic aspects that need to be considered in their design [6]. However, compared to other design disciplines, such as industrial design or graphic design, architecture has done little to clarify its aesthetic epistemology or to empirically test its aesthetic principles [8]. Therefore, there is a need for more research on how consumers perceive and prefer different architectural products and what factors influence their aesthetic judgments.

One aspect of architectural products that has received little attention in the literature is the design of timber joints. Timber joints are structural elements that connect two or more pieces of wood together to form a rigid frame or assembly. Timber joints have been used for centuries in various types of buildings and structures, such as houses, bridges, or pagodas [9]. Timber joints have both functional and aesthetic roles in architectural

products: they provide strength, stability, durability, and flexibility to the structure; they also express craftsmanship, creativity, innovation, and harmony with the natural environment [9]. However, there is a lack of empirical evidence on how consumers perceive and prefer different types of timber joints and what visual appearance attributes affect their aesthetic preference.

The aim of this study is to fill this gap by investigating the aesthetic preference of consumers for different types of timber joints used in an architectural product. The objectives of our study were threefold: firstly, to explore how Australian respondents rate various types of timber joints based on visual appearance attributes; secondly, to examine whether significant differences in aesthetic preferences exist among respondents based on sociodemographic characteristics; and thirdly, to utilise the findings to inform design decisions for a pagoda in Melbourne.

We conducted an online survey with 114 Australian participants, who rated five timber joints on seven visual appearance attributes. We also examined the influence of sociodemographic factors on their aesthetic preference. Several studies have also shown that the aesthetic impact depends on, to a substantial extent, the different sociodemographic factors, including age, class, social status, health, wealth, and so on [10]. Therefore, the research question for this study is "How does the visual appearance of timber joints affect the aesthetic preference of Australian consumers with different sociodemographic characteristics?" This research question is important because it explores how visual appearance affects aesthetic preference in a novel context of architectural products and timber joints, which has not been extensively studied before. It also contributes to the understanding of how sociodemographic factors shape aesthetic judgments and preferences, which has implications for design practice and theory. The study was conducted in the context of designing a pagoda for a cemetery in Melbourne. The results of the study were used to assist in design decisions for building the pagoda.

The study's insights into the aesthetic preferences of Australian consumers for timber joints present an opportunity to bridge aesthetics with sustainability. Timber, as a sustainable material, can be harnessed not only for its functional and visual appeal but also for its environmental benefits. Designing timber joints that resonate with consumers' aesthetic sensibilities while adhering to sustainability principles can contribute to a more eco-conscious consumer culture, where the visual appeal is intertwined with responsible material choices and production methods. This holistic consideration reinforces the idea that beauty is not merely a superficial quality, but an intrinsic characteristic deeply connected to the product's broader environmental footprint. Ultimately, this forward-looking approach seeks to redefine the relationship between aesthetics and sustainability, fostering a consumer culture where the visual attractiveness of a product is inseparable from its ethical and environmental considerations. By prioritising both aesthetic preferences and sustainability in the design of timber joints, designers and consumers alike can actively contribute to a more conscientious and environmentally friendly marketplace. This paradigm shift represents not only a response to current environmental concerns but also a proactive stance that considers the broader implications of design choices on the planet's well-being.

## 2. Literature Review

### 2.1. Aesthetics Context

Aesthetics is omnipresent in all aspects of our life. A large amount of literature is available defining aesthetics. However, its definition differs over thousands of years [11–13], concerning aesthetic experiences, preferences, response, development, or education. Conflicting statements occur mainly from how the terms are construed and applied in discussions and studies or how broadly or precisely the terms are delineated. For example, a broader definition of aesthetics is "the awareness and appreciation of pleasant sensory experiences" [14], and "aesthetics is asking questions and searching for awareness about the nature of art" [15]. A narrower definition of aesthetics is "the ability to critically evaluate works of art according to criteria defined by culture" [14]. Aesthetics can represent

extensive capabilities and responses ranging from arts and science, while Hegel refers to aesthetics as the science of sensation, of feeling [16]. However, in contrast to literature and painting, design disciplines have done little to clarify aesthetic epistemology [8].

An older definition of aesthetics pointed out that aesthetics is a living and concrete experience formality, which shows that aesthetic experiential consequences should be the standard of aesthetic evaluation [17]. Hence, aesthetic experience is often defined as a subjective pleasuring experience toward objects. Aesthetics is the subjective inner experience, judgment, and evaluation of humans with particular aesthetic abilities and thoughts resulting from objects that are commonly seen as beautiful and aesthetic, as well as the state of mind with pleasure and peace [3]. Generally, most authors' accepted definition of aesthetics encompasses the capability to understand, react, and be sympathetic to the natural environment and human creations.

In addition to explaining the wide-ranging term "aesthetics", the literature also pursues to define its associated facets. For instance, "aesthetic scanning" or the procedure and "aesthetic response" or the outcome are quite common in the literature review [18]. The former is defined as what is seen when closely looking at an artwork [19], while the latter is a broader interpretation focusing on the object's attributes, meaning, and artistic purposes [20]. A broader definition is given as the aesthetic response is a distinct kind of response that deals with feelings and talks about feelings [21]. Therefore, encompassing aesthetic response is an aesthetic preference that could be different due to personality, emotion, social–cultural experience, goal, and expectation. For instance, a positive review often comes with a good mood. Individual attributes have a certain influence on aesthetic response [22]. The most effective individual attribute is highly related to culture and educational background [23].

There is a link between the aesthetic preference of an individual with the environment as the link can affect emotion, physical response, and behaviour [7]. An individual's affective judgement can be based on whether that person likes or loathes a particular environment, whereas the emotional reaction may be instigated by pleasure or arousal associated with that environment [24]. We find conflicting ideas or concepts of objectivism and subjectivism in aesthetic preference literature. The former denotes that aesthetic quality is inherent with object property; therefore, the aesthetic preference of an individual is directed by the physical outcome of that object, while the latter entirely rests on the individual [25]. In the latter case, the aesthetic value does not depend on the intrinsic quality of objects but on how an individual interprets them with variable amounts of learned information [26].

### 2.2. Aesthetic Pleasure and Aesthetic Interest

Although research on aesthetics is directed primarily concerned with artworks, any object can be aesthetically valued and is often designed to bring aesthetic pleasure [27]. Although there are arguments on defining aesthetic pleasure, the three main views to define aesthetic pleasure are objectivist, subjectivist, and interactionist [28]. For some, aesthetic pleasure is based on an object's innate properties (i.e., symmetry, balance, proportion, and complexity) that cause pleasure. According to the subjectivist standpoint, an object is aesthetically pleasing if that object pleases our senses, so beauty becomes an act of the observer's individuality [29,30]. As stated by Blijlevens et al. [28], "aesthetic pleasure results from both the objective properties of an object and the perceiver's characteristics, i.e., aesthetic pleasure is a consequence of how perceivers and objects relate. Thus, aesthetic pleasure is value positive intrinsic and objectified".

Although aesthetic pleasure and aesthetic interest are considered positive responses in aesthetic preference research [31], the findings are contradictory. For example, many studies report that aesthetic preference may be prompted by the processing eloquence of an object by the observer, while others find that aesthetic liking is influenced by the complexity or stimulus novelty of the object, which is difficult for the observer to process [32].

### 2.3. Visual Appearance and Aesthetic Preference

The first empirical research on people's aesthetic preference for lines, forms, colours, and shapes can be traced back to Fechner's research [33]. Since then, researchers have tried various ways to explore people's aesthetic preferences for shapes and objects related to product design.

In earlier studies on the design element "line", it was suggested that curved lines are more beautiful, graceful, and pliable than straight lines [34]. This notion was also endorsed by another study that explored curved lines as gentle, merry, and quiet and angled lines as serious, agitating, and hard [35]. More recent studies with line drawings illustrate that curved lines can be associated with a quieter affective state while angular lines with an active state [33,36]. Most studies with two-dimensional shapes illustrate that asymmetry, low contrast, and angularity are less preferred compared to symmetry, high contrast, and smoothness [37].

A study with shapes revealed that circles, spheres, and curves represent softness, love, and warmth [38]. However, the study revealed some inconsistency in the association of physical form by the respondents. For example, circles, spheres, angular shapes, and straight shapes were correlated with fast instead of slow. We also find evidence that curvature was studied not only in product design but also in communication design. For example, a study on typography shows that round letters were considered more pleasant than angular letters [39]. A more recent study also reveals a preference for rounded shapes and typefaces over angular shapes and typefaces [40]. A study of complex applied stimuli found an aesthetic advantage for rounded car designs, which were perceived to be more attractive [41]. A later study on the same also supports this fact [42]. However, other studies [31,43,44] found a counter to the preference for a rounded shape. Therefore, it can be argued that the relationship between the level of curvature and viewers' preferences may be affected by other factors, such as other physical properties or contextual factors [45].

There is empirical support for the preference for rounded shapes but there are inconsistencies in the findings. Differences in the association are more pronounced for closed shapes (e.g., circles) than lines (curves). That is why the evidence is mixed when considering more complex and applied stimuli. Factors such as typicality, familiarity, and congruity may also play an essential role in this regard [46]. Though past research provides some support for the effect of angularity on preference, two factors (typicality and symmetry) may have confounded the effect of angularity [33]. Individual differences and the role of expertise have become an interesting outcome of some research [47], which suggests that experts are more sensitive to historical and compositional features rather than properties of stimuli.

Many researchers [48–50] in recent times have tried to understand consumer perception of product appearance. They all acknowledge that the first impression of a product's appearance can have a comprehensive effect in determining a user's attitude toward the product and can affect cognitive, affective, and behavioural responses to the product [51]. This, in turn, can influence users' interest in such a product [6]. Research confirmed that people naturally find beautiful things attractive [52]. However, a user's personality and socio–cultural status may influence judgments of aesthetic value [53].

### 2.4. Aesthetics and Product Design

According to Bloch et al. [54], "product design is a broad term comprising a substantial range of engineering-related attributes such as ergonomics, production efficiency, strength, recyclability, and aesthetics". By varying different aspects of product appearance, including form, material, and colour, designers try to communicate messages and obtain consumers' responses [6]. Since market success is largely dependent on the product form [2], a successful design satisfies functional requirements and is aesthetically pleasing to the user [39,55,56]. Research indicates that culture and current fashion may influence users' responses to product designs [46]. Design features can be systematically identified, which will increase users' aesthetic pleasure [57]. Many researchers [4,58] also stressed the

importance of the best combination of typicality and novelty to achieve higher aesthetic preference among consumers.

### 2.5. Timber Joints

Timber joints play a crucial role in connecting individual timber elements within architectural structures, effectively transferring loads and ensuring structural integrity. The historical use of timber in construction has emphasised the significance of designing and implementing these joints. As highlighted by Yang et al. [59], timber joints provide strength and stability to the overall building, evolving over time under the influence of regional traditions, material, and technological advancements, and the need for efficient construction methods [60]. Traditional timber joints, such as mortise and tenon joints, have a rich history in Asian countries, preserving the original beauty and aesthetics of timber without relying on steel fasteners [61]. However, advancements in fasteners and joint design have revolutionised timber construction, enabling the efficient construction of large-scale structures with enhanced durability and performance [62]. The behaviour of timber joints is intricately linked not only to the load-carrying capacity of individual fasteners but also to their form and interaction within the joint itself [63]. Various studies, including assessments of birdsmouth connections, rounded dovetail connections, mortise and tenon connections, dowel-type connections, and wood-pegged timber frames, were conducted to understand the mechanical behaviour of traditional timber joints. Additionally, researchers have explored joint strength through computer simulations [64–66].

In recent years, there has been a growing interest in utilising timber as a primary structural material for multi-storey residential and commercial buildings [67]. This shift is primarily driven by an increased awareness of the sustainability benefits associated with timber construction. Studies [68,69] indicate that timber structures exhibit lower levels of embodied carbon compared to materials like concrete or steel. However, a key challenge in timber construction lies in the design and implementation of timber joints, crucial for ensuring the structural integrity and stability of the overall timber structure [59]. Consequently, the use of timber joints has gained significance in the construction of multi-storey and long-span structures, contributing to resistance, ductility, and energy dissipation within the overall framework.

Timber joints, undergoing evolution shaped by traditions and advancements, hold a pivotal role in connecting elements within architectural structures. The design intricacies of these joints not only contribute to structural integrity but also play a key role in influencing the aesthetic preferences of individuals. Despite the sustainability benefits of timber in multi-storey buildings, the challenge lies in designing joints that not only meet structural requirements but also align with aesthetic preferences, fostering a harmonious integration of form and function within the overall architectural design. The literature review sets the stage for the research question which aligns with the discussion on visual appearance's role in aesthetic preference and the complexities surrounding factors like curvature and cultural influences. It also considers the subjectivity of aesthetic responses and the potential variation in preferences among individuals with different sociodemographic backgrounds.

## 3. Method

### 3.1. Participants

The survey was available to any Australian 18 years of age and above. Participants with restricted or no capacity or authority to provide voluntary and informed consent and under 18 years of age were prohibited from the survey. Participants were recruited from Australia by Qualtrics©, a skilled management platform that selects participants based on specifications established by the researchers. A total of 114 participants were actively recruited from various regions of Australia, with a careful selection process helping ensure that the survey's participant pool accurately represented the diverse sociodemographic characteristics of the Australian population.

### 3.2. Procedures

The research survey was conducted in an online format, aligning with the ethical guidelines outlined in the National Statement on Ethical Conduct in Human Research. Adhering to these ethical principles, the study ensured that every participant was fully informed about the survey's objectives, and their consent to participate was implied based on their voluntary engagement. To safeguard the privacy and confidentiality of the participants, a coding system was applied to all collected data, effectively anonymising the information. The survey itself was comprised of two key components. Firstly, it encompassed nine demographic questions aimed at gathering background information about the participants. These questions provided valuable context about the individuals taking part in the study. Secondly, the survey examined the aesthetic aspects of timber joints, presenting participants with a visual exploration of five different timber joints used in the construction of a pagoda (as depicted in Figure 1).

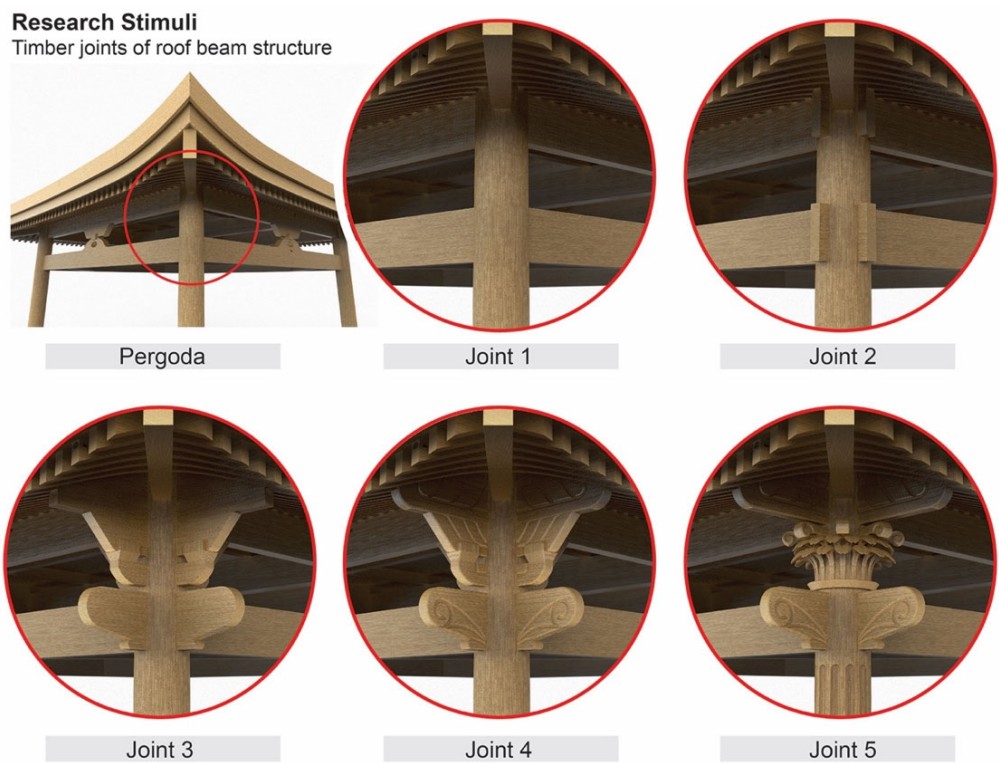

**Figure 1.** The five types of timber joints used as stimuli for this research.

For each of these images, seven distinct visual appearance questions were posed. Participants were encouraged to express their opinions and preferences on a seven-point Likert scale, spanning from "disagree" to "agree". This comprehensive approach allowed for a nuanced understanding of participants' sentiments and judgments regarding the visual appeal of timber joints. The primary focus of this research revolved around the evaluation of participants' responses to these visual appearance questions, specifically in the context of aesthetic preference. Through a careful analysis of these responses, the study aimed to unravel deeper insights into the factors influencing individuals' preferences for timber joints in architectural products. This methodical approach not only adheres to ethical standards but also ensures the richness and depth of the gathered data, contributing to a more robust understanding of consumer perceptions in the realm of architectural aesthetics.

### 3.3. Stimuli

The experimental elements in this study comprised visual representations of five distinct timber joints, each depicting a detailed connection between a post and a roof beam for

a pagoda (refer to Figure 1). These computer-generated images served a dual purpose: to gauge public preferences and to provide valuable insights for the design and construction of a pagoda intended for a cemetery in Melbourne. These stimuli exhibited a deliberate variation in design complexity, ranging from a simpler and more straightforward configuration, denoted as "Joint 1", to a more intricate and ornate arrangement, represented by "Joint 5". Joint 1 leaned towards a more angular aesthetic, while Joint 5 featured curvaceous elements. Importantly, participants were not provided with any information related to performance characteristics or other product features. This was carried out to ensure that participants' responses were solely influenced by their aesthetic preferences. To maintain consistency and minimise potential confounding variables, all stimuli were captured from the same angle and setting. This standardisation allowed for a fair and unbiased comparison of the timber joints. While certain aspects of the experimental environment, such as lighting and perspective, were carefully controlled to ensure uniformity across all stimuli, it is important to note that certain visual characteristics, inherent to the designed items intended to resemble real-world structures, remained beyond experimental control. This inherent variability is a natural part of working with designed items meant to emulate realism.

*3.4. Variables*

Since the research question is: "How does the visual appearance of timber joints affect the aesthetic preference of Australian consumers with different sociodemographic characteristics?", the independent variables are the sociodemographic characteristics of the respondents, such as age, gender, education, ethnicity, employment, and occupation. These are the factors that are assumed to influence the aesthetic preference of timber joints. The dependent variable is the aesthetic preference of Australian consumers, which was measured by their ratings on seven visual appearance questions that measure the aesthetic preference of timber joints, such as whether the joint appears strong, functional, difficult to manufacture, long-lasting, innovative and appropriate for the overall structure. These are the outcomes that are expected to vary depending on the independent variables. The mediating variable is the object's visual appearance, which is operationalised by the level of curvature of the timber joints. This is the factor that is hypothesised to explain how the independent variables affect the dependent variables. In other words, the sociodemographic characteristics of the respondents may influence their perception of the visual appearance of the timber joints, which in turn may influence their aesthetic preference.

## 4. Statistical Analysis

The major objective of data analysis in this study was to investigate how the visual appearance of timber joints affects the aesthetic preference of Australian consumers with different sociodemographic characteristics. The SPSS (Statistical Package for the Social Sciences) version 28 program was used for data analysis. Before commencing data analysis, data were checked for possible errors. Furthermore, all assumptions of the statistical method applied for this study were validated and met before any analysis. Due to the nature of the investigation, the one-way ANOVA was employed for this purpose. A one-way analysis of variance is used when the data are divided into groups according to only one factor. The questions of interest are, "Is there a significant difference between the groups? And, if so, which groups are significantly different from which others?" As the question had a substantial similarity with the above, a one-way analysis of variance was used to find the answer to the question. The process includes testing the assumptions, presenting the results, and interpreting the findings.

Checking the assumptions of ANOVA: The analysis results may be incorrect or misleading due to a violation of one or more of the one-way ANOVA test assumptions. Therefore, data were tested before proceeding with the analysis to ratify the assumptions.

- Lack of independence: the sample population was independent, and respondents were unaware of who else responded to the online survey to avoid correlated samples.

- Normal distribution: The sample size (114) was considerably above the suggested sample size (30+) [70]. Equal-interval scale was used to measure the dependent variables.
- Homogeneity of variance: to examine whether each group's variability was not considerably different, Levene's test for equality of variance was conducted and demonstrated acceptable.
- Type I error: the alpha level was selected as 0.05, which ensured a minimum type 1 error.

## 5. Results

In this section, we explore timber joint aesthetics in architectural products, sharing the key findings from our survey. We begin by introducing the survey participants and highlighting their sociodemographic characteristics. We then move on to the findings of the inferential statistics, presented in two sections. The first section validates statistical assumptions, while the second section interprets the analysis results, shedding light on the relationship between timber joints and architectural aesthetics.

### 5.1. Descriptive Statistics

In our survey analysis, we organised respondents into three distinct age groups: "Young" (ages 18–34), "Middle-aged" (ages 35–59), and "Older adults" (60 years and above). The demographic composition of our sample population predominantly featured individuals with Australian or indigenous heritage, with a notable presence of young females holding at least a bachelor's degree and actively engaged in full-time employment. Despite the absence of a dominant occupational category among respondents, a considerable portion lacked direct affiliations with the timber, construction, or design sectors. To enhance clarity and streamline our analysis, we merged certain occupational groups due to their low representation. For example, the "others" category now includes a diverse mix of individuals engaged in work-from-home arrangements, retirees, and stay-at-home parents. In order to visually represent the distribution of our respondents, we thoughtfully included Figures 2 and 3. These figures serve to illustrate the percentages within their respective age and employment categories, providing a more vivid and comprehensible presentation of our survey's demographic composition. This thorough categorisation not only enriches the understanding of our sample but also contributes to the depth and reliability of our findings.

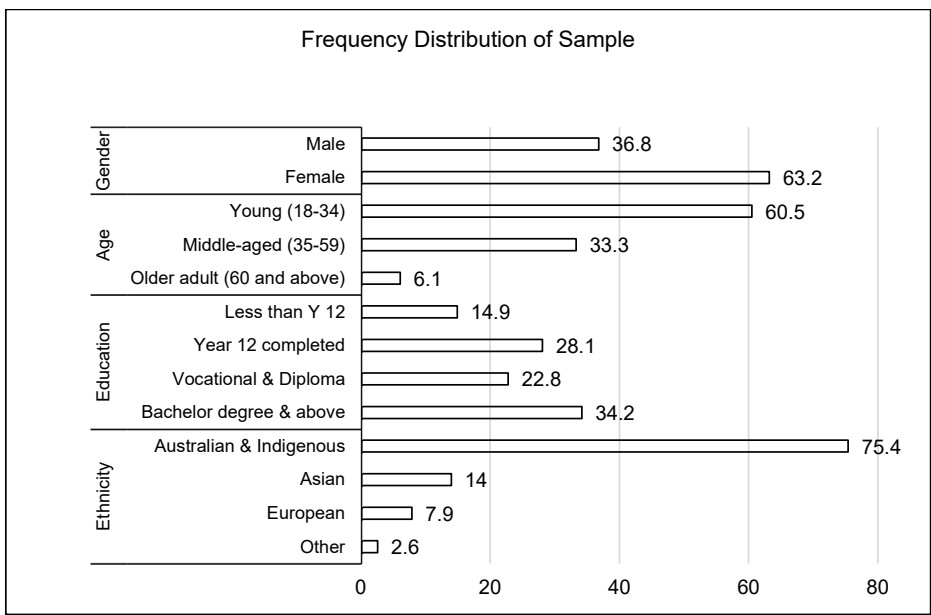

**Figure 2.** Characteristics of the respondents: gender, age, education, and ethnicity.

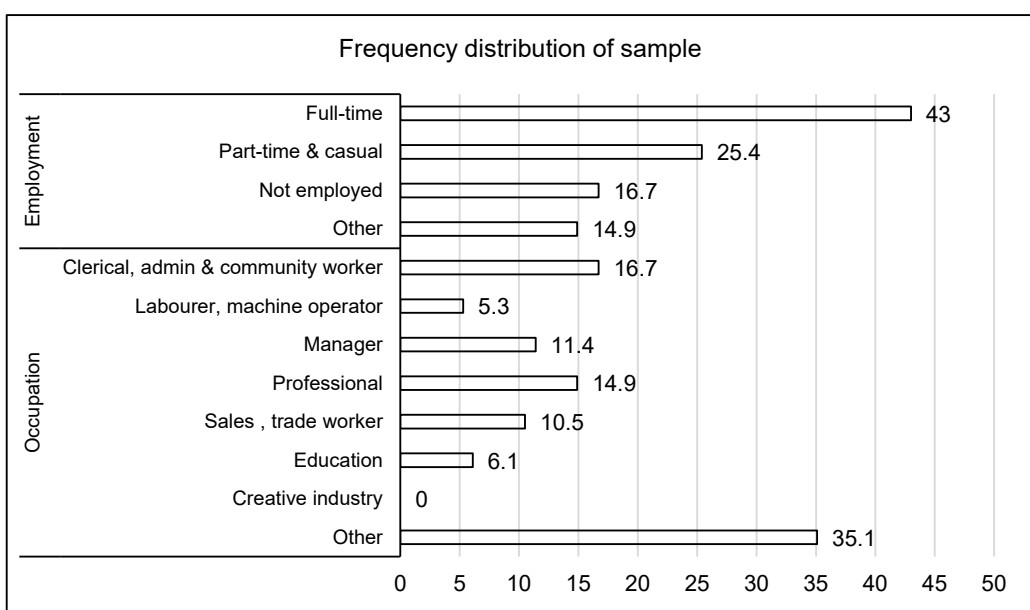

**Figure 3.** Characteristics of the respondents: employment and occupation.

*5.2. Overall Aesthetic Preference*

The "Overall Aesthetic Preference" section summarises the main findings of the survey regarding the respondents' ratings of the five timber joints on the statement "Overall, the joint is aesthetically pleasing". The section highlights the following points.

Among the respondents, more than 71 percent agreed that Joints 1, 3, 4, and 5 possessed aesthetic appeal. Joint 1 and Joint 5 were the most preferred joints by all employment categories. Joint 1 was characterised by simplicity and angularity, while Joint 5 featured intricate curvature. This suggests that the visual appearance of the joints is a key determinant of their aesthetic appeal. Joint 2 was the least preferred joint by all employment categories. Joint 2 was a simple and straight joint that lacked visual interest and innovation. This suggests that the respondents valued complexity and novelty in their aesthetic preferences. Part-time and casual workers and the unemployed exhibited higher preference scores for all joints, while full-time employed respondents scored lower. This suggests that employment status plays a role in how individuals evaluate the visual aesthetics of timber joints, with employed individuals exhibiting a more discerning approach.

*5.3. Inferential Statistics*

The outcomes of the one-way ANOVA analysis present a notable insight: sociodemographic factors such as age, gender, education level, ethnicity, and occupation exhibited limited to no significant impact on individuals' aesthetic preferences for timber joints. Surprisingly, the majority of sociodemographic variables, encompassing age, gender, educational background, ethnicity, and occupation, did not seem to play a discernible role in shaping how individuals assessed the aesthetic appeal of timber joints. However, a noteworthy exception emerged in the form of employment status. Contrary to the other variables, the data suggested that employment status did influence these preferences. In essence, whether an individual was employed or not appeared to affect their aesthetic preferences in the realm of timber joints. While the precise nature of this influence warrants further investigation, it implies that employment status may be a key factor in elucidating why certain individuals favoured particular aesthetic attributes of timber joints over others. Consequently, the subsequent results shown in Figure 4 are contingent solely on respondents' employment status.

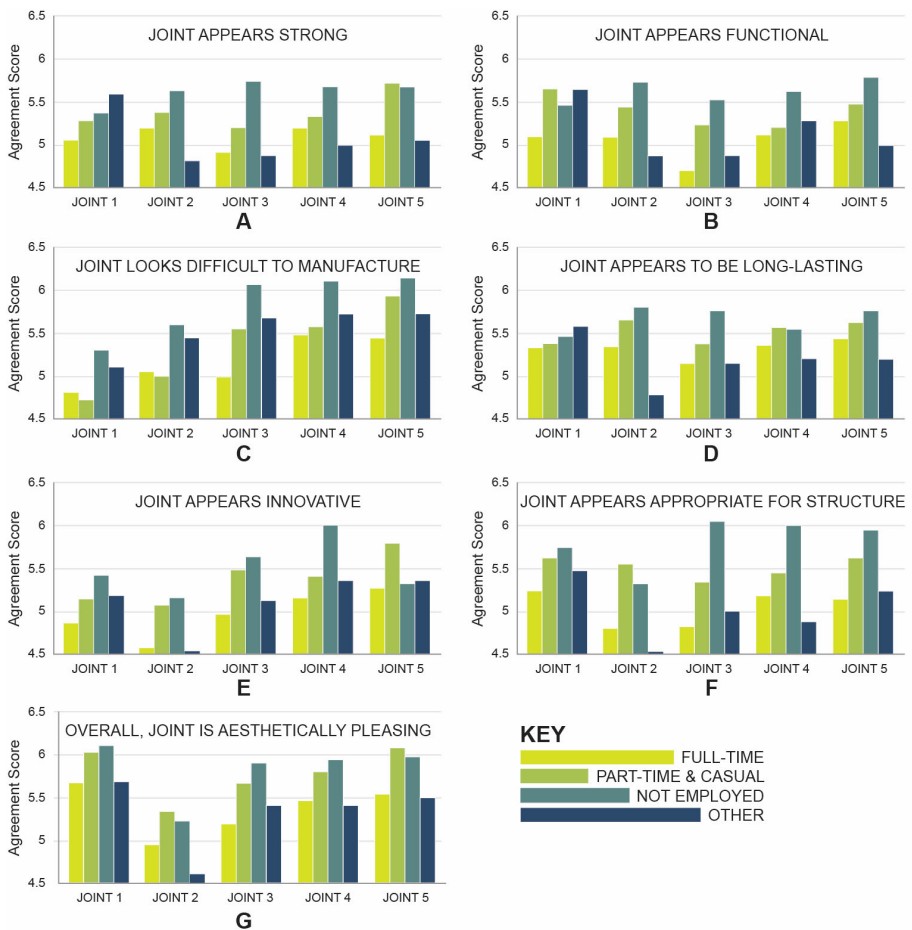

**Figure 4.** Mean agreement scores for the visual appearance of all timber joints by employment. Subfigures (**A**–**G**) show individual results for the seven distinct visual appearance questions.

Joint Strength Perception: Joint 1 received unanimous agreement regarding its perceived strength from all employment groups. Particularly, non-employed respondents showed relatively high agreement across all joint types (Figure 4A). The results imply that the visual appearance of timber joints plays a significant role in how their strength is perceived. This could be attributed to the visual stability these joints convey, possibly associated with notions of strength and durability. This finding aligns with our research question, emphasising the crucial role of visual aesthetics in shaping the aesthetic preferences of Australian consumers.

Functional Aspect: In terms of functionality, Joint 1, distinguished by its angularity, and Joint 5, recognised for its curvature, stood out as more functional compared to other joints. This suggests that their forms are perceived as more practical or useful. Joint 1 was notably favoured by part-time and casual workers, as well as individuals categorised as "others," while Joint 5 found preference among part-time and casual workers and the unemployed. Intriguingly, unemployed respondents consistently showed high agreement scores irrespective of the joint type (Figure 4B).

The preference for specific joints based on their angular or curved nature relates directly to the research question, as it highlights that the visual features of timber joints significantly impact how individuals perceive their functionality. This effect remains consistent across different employment categories.

Manufacturability Impression: The visual appearance of the joints significantly influenced perceptions of difficulty in manufacturing. The data revealed a gradual progression of perceived difficulty, with Joint 1 seen as less challenging to manufacture and Joint

5 regarded as the most challenging. This could reflect an appreciation for the skill and craftsmanship required to create more complex forms.

Among full-time employed respondents, all joints appeared less difficult to manufacture, contrasting with the opposite trend observed among the unemployed. Notably, there was a statistically significant difference in the mean agreement score for Joint 3 between full-time and unemployed respondents, a finding reinforced by practical significance confirmed through effect size analysis (Figure 4C). The impact of visual appearance on the perception of manufacturing difficulty highlights the role of aesthetics in shaping preferences. The clear preferences of full-time employed and unemployed respondents for specific joints emphasise the interaction between employment status and aesthetic judgments.

Longevity Perception: Joint 5 received relatively higher agreement scores from all employment categories when it came to being perceived as long-lasting, which could be due to its robust and durable appearance. Joint 1 also scored well in this aspect. Notably, respondents in the "other" employment category rated Joint 2 lower, while unemployed respondents consistently provided high ratings for all joints (Figure 4D). The findings regarding the perceived longevity of joints illustrate that visual cues play a substantial role in determining how consumers assess the durability of timber joints. This observation aligns with our research question by emphasising the connection between visual aesthetics and preferences, with variations across employment categories.

Innovation Aspect: Joint 2 was perceived as less innovative compared to other joints. In contrast, Joint 4 received the highest ratings, indicating that unique and unconventional forms can be particularly appealing, especially to unemployed respondents. Both part-time and casual workers and the unemployed consistently awarded higher scores compared to other employment categories for the innovation aspect (Figure 4E). This highlights that the visual characteristics of timber joints strongly influence the perception of innovation, emphasising the significant impact of joint appearance on aesthetic preferences, especially among part-time and casual workers and the unemployed.

Suitability for Overall Structure: Joint 1 was identified as the joint most suitable for the overall structure, indicating the significance of harmony and coherence in aesthetic preference, closely followed by Joint 5. Full-time employed respondents tended to have more stringent opinions regarding the appropriateness of the joints, while non-employed respondents expressed relatively high levels of agreement (Figure 4F). The perception of Joint 1 and Joint 5 as most suitable for the overall structure points out the impact of visual aesthetics on preferences. The variations in agreement scores between full-time employed and non-employed respondents further underscore the influence of employment status on shaping these judgments.

Overall Aesthetic Appeal: Notably, the scores for overall aesthetic preference indicated that Joint 1, characterised by simplicity and angularity, and Joint 5, featuring intricate curvature, consistently ranked as the most aesthetically pleasing joints across all employment categories. This suggests that a harmonious balance between strength, functionality, manufacturability, longevity, innovation, and suitability contributes to aesthetic appeal. Except for Joint 2, all other joints also received higher ratings. Interestingly, there was a significant decrease in preference for Joint 2 among individuals working from home, stay-at-home parents, and retired respondents. In general, part-time and casual workers and the unemployed exhibited higher preference scores, while full-time employed respondents scored lower (Figure 4G).

The observation that Joint 1 and Joint 5 are consistently perceived as the most aesthetically pleasing joints aligns perfectly with the research question. It suggests that the visual appearance of timber joints is a key determinant of aesthetic preferences among Australian consumers. Moreover, the varying preferences across employment categories reinforce the importance of considering sociodemographic factors, such as employment status, in understanding these preferences.

A one-way ANOVA analysis shows that people's aesthetic preferences for timber joints are mainly influenced by how the joints look, rather than by their age, gender,

education, ethnicity, or occupation, except when it comes to their employment status. But it is important to note that the authors did find one interesting statistical result regarding Joint 3, which is explained later.

Manufacturability Impression: As depicted in Table 1, a statistically significant difference was observed at the $p < 0.05$ level among the four employment groups concerning the perception of the difficulty in manufacturing Joint 3. This finding prompted a detailed examination, and Table 2 presents the outcomes of our post-hoc comparisons utilising the Tukey HSD test. These comparisons identified specific differences in mean scores between the employment groups. Particularly noteworthy is the significant difference in perceptions of Joint 3's manufacturability between Group 1 (full-time employed) and Group 3 (not employed). However, Group 2 (part-time and casual) and Group 4 (other) did not exhibit significant differences in comparison to either Group 1 or Group 3. For a clearer understanding of these mean differences, we visually represented them in Figure 5. Additionally, we calculated the effect size, denoted by Eta squared, which equated to 0.103. In widely accepted terms, this falls within the medium effect range [71]. Essentially, this indicates that approximately 10.3 percent of the variability in scores related to the perception of "Joint 3 looking difficult to manufacture" can be attributed to the respondents' employment status. These findings suggest that while employment status notably influences how individuals perceive the manufacturing complexity of Joint 3, it is just one among several factors contributing to these perceptions. Further exploration may help us uncover the nuanced dynamics at work within this relationship.

**Table 1.** Significance of employment status determining manufacturing difficulty of Joint 3.

| ANOVA | | | | |
|---|---|---|---|---|
| **Please Rate Your Level of Agreement on Visual Appearance of Timber Joint 3—The Joint Looks Difficult to Manufacture** | | | | |
| | **Sum of Squares** | **df** | **Mean Square** | **F** | **Sig.** |
| Between Groups | 28.732 | 3 | 9.577 | 4.231 | 0.007 |
| Within Groups | 249.022 | 110 | 2.264 | | |
| Total | 277.754 | 113 | | | |

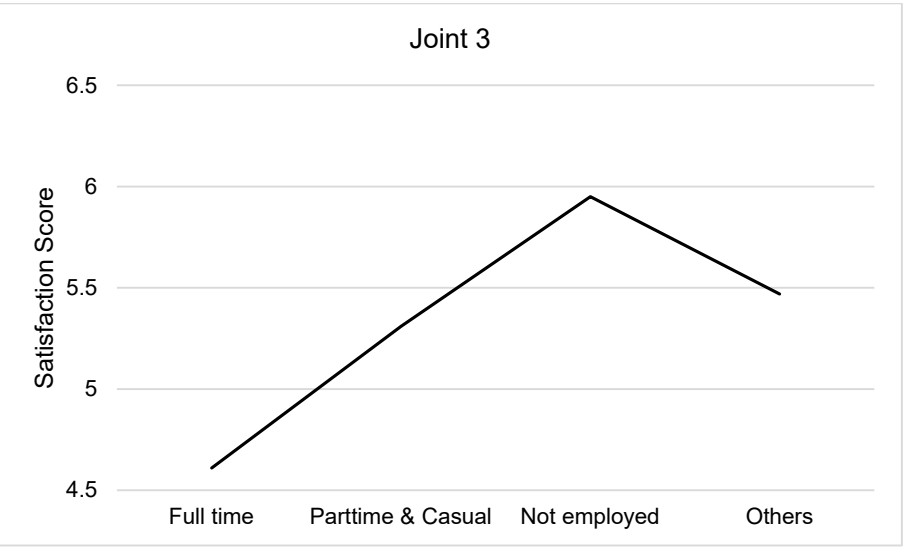

**Figure 5.** Agreement scores of Joint 3 by employment status: the joint looks difficult to manufacture.

**Table 2.** Post-hoc test of employment status on manufacturing difficulty of Joint 3 showing significant mean differences in group scores.

| Multiple Comparisons | | | | | | |
|---|---|---|---|---|---|---|
| Dependent Variable: Please Rate Your Level of Agreement on the Visual Appearance of Timber Joint 3—The Joint Looks Difficult to Manufacture | | | | | | |
| Tukey HSD | | | | | | |
| (I) Employment | (J) Employment | Mean Difference (I − J) | Std. Error | Sig. | 95% Confidence Interval | |
| | | | | | Lower Bound | Upper Bound |
| 1.00 | 2.00 | −0.698 | 0.353 | 0.202 | −1.62 | 0.22 |
| | 3.00 | −1.335 * | 0.407 | 0.007 | −2.40 | −0.27 |
| | 4.00 | −0.858 | 0.424 | 0.185 | −1.96 | 0.25 |
| 2.00 | 1.00 | 0.698 | 0.353 | 0.202 | −0.22 | 1.62 |
| | 3.00 | −0.637 | 0.444 | 0.481 | −1.80 | 0.52 |
| | 4.00 | −0.160 | 0.460 | 0.985 | −1.36 | 1.04 |
| 3.00 | 1.00 | 1.335 * | 0.407 | 0.007 | 0.27 | 2.40 |
| | 2.00 | 0.637 | 0.444 | 0.481 | −0.52 | 1.80 |
| | 4.00 | 0.477 | 0.502 | 0.778 | −0.83 | 1.79 |
| 4.00 | 1.00 | 0.858 | 0.424 | 0.185 | −0.25 | 1.96 |
| | 2.00 | 0.160 | 0.460 | 0.985 | −1.04 | 1.36 |
| | 3.00 | −0.477 | 0.502 | 0.778 | −1.79 | 0.83 |

* The mean difference is significant at the 0.05 level.

### 5.4. Summary of Findings

In summary, our research findings can be distilled into several key points and expand on previous work from the authors [72]:

- Visual appearance dominates aesthetic preferences: The results of our one-way ANOVA analysis highlight that the visual appearance of timber joints plays a pivotal role in shaping people's aesthetic preferences. In essence, how the joints look is a central determinant of their aesthetic appeal.

- Employment status significantly impacts perception: The impact of sociodemographic factors on aesthetic preferences is however not uniform. Notably, employment status stands out as a sociodemographic variable that exerts a significant influence. This influence is particularly pronounced in the case of Joint 3, where respondents' employment status has a statistically significant impact on their perception of the joint's appropriateness for the overall structure and its level of difficulty in manufacturing.

- Practical significance confirmed: The effect size analysis underscores the practical significance of the differences in respondents' perceptions of Joint 3. It verifies that these variations in responses regarding the appropriateness and manufacturability of the joint are not merely statistically significant but also hold practical importance.

- Consistent mean value patterns: Interestingly, when examining the mean values of respondents' perceptions concerning timber joints' visual appearance, we observe a consistent pattern. Respondents with full-time employment, the largest group comprising 43 percent of the sample, tend to provide more stringent and specific responses compared to those who are not employed. This pattern suggests that employment status plays a role in how individuals evaluate the visual aesthetics of timber joints, with employed individuals exhibiting a more discerning approach.

In essence, our research demonstrates that the visual appeal of timber joints is a primary driver of aesthetic preferences, cutting across various sociodemographic characteristics. However, employment status emerges as a distinct factor, particularly influencing

perceptions of Joint 3's suitability for the overall structure and manufacturability. This nuanced interplay between employment status and aesthetic judgments enriches our understanding of how individuals with different sociodemographic backgrounds evaluate timber joints.

## 6. Discussion

This study explored the aesthetic preference of Australian consumers for different types of timber joints in a pagoda design. The study used an online survey with visual stimuli and measured the respondents' ratings on seven visual appearance attributes. The study also examined the influence of sociodemographic factors, such as age, gender, education, ethnicity, employment and occupation, on aesthetic preference. The main findings of the study are:

Employment Status significantly impacts perception: Employment status is a complex factor influenced by various sociocultural and individual aspects and can significantly influence an individual's perspective, preferences, and emotional responses. It also plays a significant role in shaping one's identity and place in society [73]. The results revealed that employment status was the only sociodemographic factor that had a significant effect on the aesthetic preference of the respondents. This finding is consistent with some previous studies that have found employment status to be a predictor of aesthetic preference [61,62]. This finding is interesting, as it suggests that employment status may be related to other factors, such as income, lifestyle, values, or exposure to design, that may affect the aesthetic judgment and preference of consumers [10,53]. The results also showed that full-time employed respondents tended to have more stringent and specific opinions on the timber joints, while non-employed respondents expressed relatively high levels of agreement for all joints. This finding implies that employment status may influence the perception of the visual appearance attributes, such as strength, functionality, difficulty, durability, innovation and suitability, of the timber joints. The most notable difference was observed for Joint 3, which was perceived as more difficult to manufacture by non-employed respondents than by full-time employed respondents. This finding may indicate that non-employed respondents have less familiarity or knowledge of the manufacturing process of timber joints, or that they have different expectations or standards for the design of architectural products [22,23,47].

Visual appearance dominates aesthetic preference: The results showed that the visual appearance of the timber joints, especially the level of curvature, was the most important factor in determining the aesthetic preference of the respondents. This finding is consistent with the literature that suggests that visual appearance is a key driver of aesthetic response and product success [1,2,27,48–50]. The respondents preferred joints that were either angular (Joint 1) or curved (Joint 5), as they perceived them as more strong, functional, long-lasting, innovative and suitable for the overall structure. This finding also supports the literature that indicates that curvature is an influential visual attribute that can elicit different emotional and cognitive reactions [34–36,38–42,45,46].

The discussion also explores some intricate relationships between respondents and their perceptions and addresses the broader question of why certain preferences emerge: Difficulty in Manufacturing Perception: Figure 6 provides visual evidence of the disparities in perception between unemployed and full-time employed respondents concerning Joint 3's manufacturability. Unemployed individuals tend to perceive this joint as more challenging to manufacture, while those in full-time employment find it less daunting. This divergence can be linked to established research showing that the visual appearance of a product can profoundly affect how consumers evaluate it [74]. Aesthetic value often becomes intertwined with visual appearance, transcending mere practical utility [75]. The intricate interplay of cultural, social, and personal factors, including personality traits, past experiences, and design expertise, can exert a significant influence on an individual's design preferences.

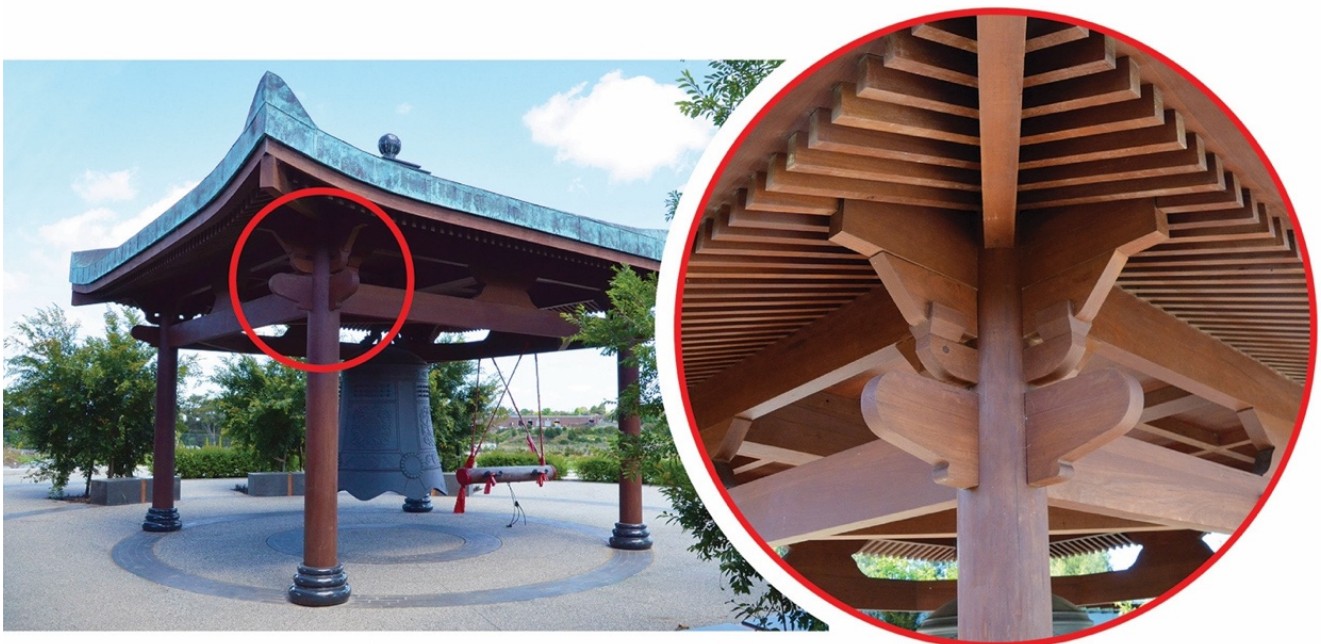

**Figure 6.** Research outcome. Manufactured pagoda located at Springvale cemetery, Melbourne, Australia. Detail of Joint 3 manufactured as an outcome of this study.

Aesthetic judgment and design insight: Furthermore, the impact of aesthetic judgment on product preference is influenced by how well a product's aesthetics align with other possessions [1]. In the case of Joint 3, full-time employed individuals may perceive its aesthetic aspects, semantic interpretations, and emotional responses as less influential. Their greater insight and experience in design may lead them to believe that Joint 3 is not as challenging to manufacture as others might think. Importantly, because the full-time employed group comprises the largest portion of our sample (43%), their opinions hold substantial weight in the joint selection process.

Complex Stimuli and Aesthetic Preferences: Previous research often focused on simple stimuli or familiar objects, often with limited sample sizes [37]. Our study advances this by examining more complex and practical stimuli, such as timber joints. The findings underscore a crucial insight: aesthetic preferences can become less consistent when dealing with intricate and real-world objects [46]. Figure 4A–G illustrate the variation in respondents' aesthetic preferences across different joint types. This variation is closely tied to the angularity and curvature of these joints, highlighting their pivotal roles in shaping evaluations [76–78].

## 7. Implementation

From the very beginning, our research aimed to gain a deep understanding of the public's preferences regarding designed objects, using timber joints as our study stimuli. Our primary motivation was to inform design decisions, particularly for a pagoda project that we, as authors, were commissioned to undertake. We employed these experimental stimuli to delve into public preferences, ranging from straightforward angular designs to more intricate, decorative curved designs.

The findings did not reveal a clear-cut preference for any specific joint design among the participants. However, a significant discovery emerged when we examined the mean scores for "Joint looks difficult to manufacture", especially in the case of Joint 3. This led us to investigate further by comparing these scores with those of other joints (i.e., 1, 2, 4, and 5). What we uncovered was intriguing: respondents who held full-time jobs consistently assigned lower scores to all joint designs compared to their not employed counterparts. This trend indicated that from the perspective of those with full-time employment, Joints

1 through 5 appeared less challenging to manufacture. Given that this group comprised the majority of responses, this statistical insight greatly influenced our decision regarding which joint to implement. It also helped dispel any concerns about negative perceptions that might have arisen with the selected joint type.

While our data did not provide a definitive preference for a specific joint type, we reviewed all results in close consultation with the manufacturer to determine the most suitable joint for production. Our focus shifted from aiming solely for the most aesthetically pleasing joint to finding one that balanced various factors. In this regard, Joint 3 emerged as a pragmatic choice, striking a compromise that resonated with numerous instances within our data. This choice was particularly apt for the overarching question, "Overall, the joint is aesthetically pleasing", where preferences were divided between Joint 1 and Joint 5. Notably, Joints 1 and 5 also garnered favour for questions related to functionality, longevity, and suitability for the overall structure. Joint 3, positioned as the middle ground, encapsulated elements from both Joint 1 and Joint 5 and, crucially, did not elicit any negative reactions from respondents. While it might not have ranked as the top favourite, it closely trailed behind and was more feasible to manufacture compared to Joint 5. Although manufacturing posed its own set of challenges, we were pleased to maintain the appealing curved contours inherent in Joint 3, which research literature had already validated as preferable to sharp contours.

Remarkably, the outcomes of our research found practical application in the production of a fully realised pagoda (see Figure 6), demonstrating the tangible impact and real-world translation of our work. While it is unlikely that every individual will be completely satisfied, the final manufactured product boasts enough qualities to please the majority, as substantiated by our extensive survey involving 114 participants.

## 8. Limitations

As with virtually any work on aesthetic preference, the results of the present study are subject to certain limitations. These limitations are summarised as follows.

The present study acknowledges the influence of typicality and novelty on aesthetic preference; however, investigating the relationship between typicality, novelty, and aesthetic preference is beyond this study's scope.

Since the study cannot claim to include all constructs and variables of aesthetic preference, the generalisability of the findings is difficult to assess except for the respondents who participated in the survey without the benefit of further research in this area.

The respondents were given the opportunity to respond only to the selected visual appearance questions. Respondents did not have the opportunity to answer an open-ended question; thus, the study excluded other aspects of their lives that may influence their aesthetic preferences.

We recognise that our study deliberately narrowed its focus, primarily centering on understanding how sociodemographic characteristics influence aesthetic preferences for timber joints. Nonetheless, we concur that taking into account the broader architectural context is crucial for practical applications in the real world.

While our findings indicate that the ethnicity of the sample population did not significantly influence the aesthetic preference for timber joints, it is important to recognise that traditional cultural factors associated with ethnicity might still play a role in participants' aesthetic judgments. These cultural influences could potentially impact preferences for specific timber joint models. We acknowledge the necessity for a more refined exploration of these cultural dimensions in future research.

While a significant effort was put towards the CAD (Solidworks 2022 version) and manufacturing data to retain an accurate manufactured outcome in relation to the stimuli, certain manufacturing details were altered due to manufacturability, material constraints, cost, and time.

## 9. Conclusions

In conclusion, our research highlights the intricate relationship between employment status and individuals' aesthetic preferences, particularly in the context of architectural elements like timber joints. While the visual appeal of these elements remains a central factor in shaping preferences, we uncovered that employment status serves as a significant sociodemographic variable that can wield considerable influence, especially in complex design scenarios like timber joints. This nuanced connection underscores the importance of considering diverse personal attributes and the visual aspects of objects when assessing aesthetic preferences. Our findings offer valuable insights for architects and designers seeking to create designs and products that resonate with a wide spectrum of consumers. However, beyond its immediate implications for design, our research also presents an opportunity to bridge aesthetics with sustainability, a critical issue in contemporary design and consumer culture.

As mentioned in previous discussions, timber, as a sustainable material, holds great potential for environmentally conscious design. The aesthetic preferences uncovered in our research can serve as a bridge to integrate sustainability principles into design choices. By crafting timber joints that align with consumers' aesthetic sensibilities, designers can simultaneously promote sustainable practices and materials. This synergy between aesthetics and sustainability reflects a forward-thinking approach, where visual appeal extends beyond surface appearances and incorporates responsible material choices and eco-friendly production methods. In essence, our research not only enriches our understanding of how sociodemographic factors influence aesthetic preferences but also underscores the potential for aesthetics to be a driving force behind sustainable design choices. By leveraging these insights, designers, manufacturers, and marketers can navigate the complex landscape of consumer preferences with precision, ultimately delivering products that resonate deeply with diverse audiences while contributing to a more sustainable future.

**Author Contributions:** Conceptualisation, B.K.; methodology, B.K. and M.M.; formal analysis, B.K. and M.M.; investigation, B.K. and M.M.; writing—original draft preparation, B.K. and M.M.; writing—review and editing, B.K. and M.M.; visualisation, B.K.; project administration, M.M. All authors have read and agreed to the published version of the manuscript.

**Funding:** This research received no external funding.

**Institutional Review Board Statement:** This study was conducted in compliance with the principles outlined in the Declaration of Helsinki and received approval from the Swinburne University of Technology Human Ethics Committee (Reference: R/2019/209, dated 18 September 2019) for research involving human participants.

**Informed Consent Statement:** Informed consent was obtained from all subjects involved in the study.

**Data Availability Statement:** The data featured in this study can be obtained upon request from the corresponding author. Please note that the data cannot be made publicly accessible due to ethical restrictions that have been granted approval.

**Acknowledgments:** The authors express their gratitude to Timberfy, an Australian manufacturer, for their collaboration with the research team in producing the pagoda for one of Melbourne's largest cemeteries. Additionally, the authors extend their thanks to the participants in the study for generously dedicating their time to complete the survey.

**Conflicts of Interest:** The authors declare no conflict of interest.

## Abbreviations

ANOVA: Analysis of Variance, a statistical method used to test differences between two or more means. SPSS: Statistical Package for the Social Sciences, a software package used for statistical analysis. $p < 0.05$: A commonly used probability value for determining statistical significance. If a result has a $p$-value less than 0.05, it is considered statistically significant. Type I error: The probability of rejecting a true null hypothesis (also known as a "false positive"). Effect size: A quantitative

measure of the magnitude of a phenomenon or effect. Post-hoc comparisons: Statistical comparisons made between group means after conducting an ANOVA test.

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
