# Peer review of "Aesthetic Preference of Timber Joints in Architectural Products"

_sustainability, doi:10.3390/su16010154_

Round 1

Reviewer 1 Report

Comments and Suggestions for Authors

The presented study is interesting and provides new insights into the aesthetic preferences for timber joints in architectural products. However, some parts of the manuscript were not adequately presented and should be rewritten. In many places, the information was repeated and sentences were similar. The abstract should be slightly modified according to the Journal's template. The Introduction combined with the Literature review section provide sufficient background information, with some changes needed. The subsections of the Material and Methods section should be rewritten as explained in the comments. The Results section should be completely rewritten because the findings are not clearly presented and the text is hard to follow, thus the significance of the obtained results is not adequately stated. The Discussion section is very poor. There are little to no discussion on similar studies. The whole Results and Discussion sections are very generalized, while the presentation of results in the form of tables and values (scores) referring to particular timber joints is not sufficient. The fact that the findings of the study were used to construct pagoda adds practical value to the study, which is very significant.

Additional comments are given in the PDF file.

Comments on the Quality of English Language

Minor editing of English language required

Author Response

We thank the reviewer for their valuable feedback. Please find our point-by-point response in the attached cover letter.

Reviewer 2 Report

Comments and Suggestions for Authors

The paper investigates consumers' aesthetic preference for different types of wooden joints used in architectural artefacts.

The treatment is exemplary for the clarity of the method and the rigorousness of the reasoning.The focus of this research revolves around the evaluation of the participants' responses. The conclusions are consistent with the evidence and arguments presented; they answer the questions posed and provide interesting reflections. There is no lack of useful self-criticism of the methods of analysis.  The results of the study were used to assist in the design decisions of an Australian pagoda. More significantly, the findings encourage sustainable architecture, focusing on the use of wood as a suitable building material to reconcile aesthetics, static and function.

Author Response

(The authors gave the same response as above.)

Reviewer 3 Report

Comments and Suggestions for Authors

This paper is interesting and shows investigate how Australian consumers’ aesthetic preferences for timber joints in architectural products are influenced by their sociodemographic characteristics and the visual appearance of the joints.

My detailed comments and recommendations related to the paper are presented below.

1. The literature review part analyzed advances related to aesthetics from different perspectives, including aesthetic context, aesthetic pleasure and aesthetic interest, visual appearance and aesthetic preference, and aesthetics and product design, but there was a lack of review on timber joints in architecture.

2.Whether the five timber joint forms chosen for this research are sufficiently representative, which includes whether these five timber joints are the most typical and include cover almost all forms of timber joints, and also whether these wood timber forms are actual or imagined.

3.As an applied study, all aesthetic considerations should be taken into account for application feasibility, it was not only focused on the aesthetic preference of timber joints themselves, but also on the individual building units, and indeed the entire complex or community.

4.The authors concluded that the independent variable of ethnicity has no effect on timber joints aesthetic preferences, but looking at the five timber joints models used in this study, some of which clearly have a Japanese Wabi-sabi aesthetic and a classical Greek Corinthian Order, it is likely that these important traditional cultural factors related to ethnicity may have an impact on the participants' judgment of aesthetics, and the authors need to take them fully into account in their study.

Author Response

(The authors gave the same response as above.)

Reviewer 4 Report

Comments and Suggestions for Authors

The paper conducted an online survey with 114 participants, who rated five timber joints on seven visual appearance attributes to study how Australian consumers’ aesthetic preferences for timber joints in architectural products are influenced by the visual appearance of the joints. They modeled 5 types of joint. SPSS software, frequency distribution, Inferential statistics  were used. The novelty is good. The revision:

1) Please add a notation list.

2) Fig. 4 needs more discussion in the text.

3) In order to provide a more comprehensive literature review on the strength of different type of joints by simulating, SPSS software, and Frequency distribution, the following relevant papers should be cited and discussed in the revised manuscript:

- Probability distribution models for the ultimate strength of tubular T/Y-joints reinforced with collar plates at room and different fire conditions. Ocean Engineering2023; 270, p.113557.

- Static strength of X-joints reinforced with collar plates subjected to brace tensile loading. Ocean Engineering2018; 161, pp.227-241.

4)  How was the model validated?  

5) line 554, “Our findings didn't” should be revised. Use of “our” should be “the” or “ the finding of the present research works” or…

6) line 539 “valuations [65, 66, 67].” Should be “valuations [65-67].”

Comments on the Quality of English Language

 line 554, “Our findings didn't” should be revised. Use of “our” should be “the” or “ the finding of the present research works” or…

line 539 “valuations [65, 66, 67].” Should be “valuations [65-67].”

Author Response

(The authors gave the same response as above.)

Reviewer 5 Report

Comments and Suggestions for Authors

The manuscript, titled Aesthetic preference of timber joints in architectural products, presents a survey for a small population based on a questionnaire.

Merits

The article is clearly written and reads well. Although the experiment was designed correctly, it's tough to use math to analyze respondents' behavior. Selection can be influenced by various factors not considered in the study. It is not known what the questionnaire was like, as its design, the order of questions, etc. Also can affect the answers given. Nevertheless, the survey seems reliable and properly interpreted.

The work is so clear and complete that I have no serious comments and believe that it can be accepted even in its present form. I only suggest considering the following points.

Suggestions

The Authors focused more extensively on the response regarding Joint 3-The joint looks difficult to manufacture. In Figure 4C, you can clearly see a large difference between the posts, which later became the subject of discussion. Figure 4F show also shows high visual differences for Joint 3, but it’s hard to determine statistical significance without test results. I suggest to consider showing all the differences for each variant. This could be in a separate table, even in a supplement, so that the reader can access all the statistics and judge the value of the study for themselves.

It is not clear whether the groups of questions were analyzed only separately or were also combined? E.g. Scores for Joints 3-4 are always high for non-employed in questions A, C and F. Combining ACF responses into a single group (e.g. strong, difficult construction) helps analyze responses between non-employed (strong responses) and full-time employed.

Details

Most references require verification and formatting consistent with MDPI references style.

https://www.mdpi.com/authors/references

Author Response

(The authors gave the same response as above.)

Round 2

Reviewer 1 Report

Comments and Suggestions for Authors

It can be seen that the manuscript was improved to a high extent according to the all Reviewers’ comments. Unfortunately, I have found that Authors have provided answers to my general comments only, which prompted me to check my first review and whether my additional comments in PDF file have been uploaded. Although at the end of my Comments and Suggestions for Authors I did stated: ‘Additional comments are given in the PDF file.’, I found out that the mentioned PDF file was not uploaded presumably due to some technical problems (I uploaded the file before submitting my review). I believe that these comments could improve the proposed Manuscript so I will upload the file again.

I apologize for any inconvenience. Thank you for understanding.

Comments on the Quality of English Language

Minor editing of English language required

Author Response

Thank you very much for sending through the marked-up PDF. Please find our rebuttal in the attached document.

Reviewer 4 Report

Comments and Suggestions for Authors

Ready to publish.

Author Response

Thank you very much for your positive response and recommending our paper for publication. Kind regards.

Round 3

Reviewer 1 Report

Comments and Suggestions for Authors

The manuscript has been improved in accordance with the suggestions, thank you.